# Physical Activity Design for Balance Rehabilitation in Children with Autism Spectrum Disorder

**DOI:** 10.3390/children9081152

**Published:** 2022-07-30

**Authors:** Andreea Maria Roșca, Ligia Rusu, Mihnea Ion Marin, Virgil Ene Voiculescu, Carmen Ene Voiculescu

**Affiliations:** 1Faculty of Physical Education and Sport, University of Craiova, 200585 Craiova, Romania; rosca.andreea.d5s@student.ucv.ro (A.M.R.); virgil.ene@anmb.ro (V.E.V.); 2Faculty of Mechanics, University of Craiova, 200585 Craiova, Romania; mihnea_marin@yahoo.com; 3Faculty of Physical Education and Sport, Ovidius University of Constant, 900470 Constanța, Romania; carmenenevoiculescu@gmail.com

**Keywords:** ASD, balance, postural control, rehabilitation

## Abstract

One of the characteristics of autism spectrum disorder (ASD) subjects is postural control deficit, which is significant when somatosensory perception is affected. This study analyzed postural stability evolution after physical therapy exercises based on balance training. The study included 28 children with ASD (average age 8 years, average weight 32.18 kg). The rehabilitation program involved performing balance exercises twice a week for three months. Subject assessment was carried out using the RSScan platform. The parameters were the surface of the confidence ellipse (A) and the length of the curve (L) described by the pressure center, which were evaluated before and after the rehabilitation program. Following data processing, we observed a significant decrease in the surface of the confidence ellipse by 92% from EV1 to EV2. Additionally, a decrease of 42% in the curve length was observed from EV1 to EV2. A t test applied to the ellipse surface showed a *p* = 0.021 and a Cohen’s coefficient of 0.8 (very large effect size). A t test applied to the length L showed *p* = 0.029 and Cohen’s coefficient of 1.27 mm. Thus, the results show a significant improvement in the two parameters. The application of the program based on physical exercise led to an improvement in the balance of children with autism under complex evaluation conditions.

## 1. Introduction

World Health Organization (WHO) data published in April 2017 show that an estimated 1 in 160 children worldwide are suffering from an autism spectrum disorder (ASD). This represents more than 7.6 million years of life adjustments depending on the disability and 0.3% of the global burden of diseases. This estimation represents an average number, and the reported prevalence substantially varies across studies. However, some well-controlled studies have reported numbers that are substantially higher. In many countries with low and medium budgets, ASD prevalence levels are still unknown. Based on epidemiological studies conducted over the last 50 years, ASD prevalence seems to be increasing globally. There are many possible explanations for this apparent increase, including greater awareness, expanded diagnostic criteria, better diagnosis instruments, and better reporting.

Recent studies suggest that autism affects about one percent of European people, accounting for more than five million people in the European Union (EU). Over the last 30 years, the number of reported autism cases has increased rapidly in all countries where prevalence studies have been performed. This increase is partly due to increased awareness of this disorder among health professionals, parents, and the general population; changes in the diagnosis criteria for autism; and children being diagnosed from an early age.

Autism is also known as an early onset disorder of the central nervous system (CNS). Even though the underlying mechanisms are still unknown, autism is usually described as a disorder of the brain since many changes in the brain have been found and analyzed [1]. In fact, ASD symptoms have been associated with ubiquitous CNS atypicality.

Autism spectrum disorders (ASDs) are a group of complex disorders involving brain development [2]. This umbrella term includes autistic disorder, Asperger disorder, and atypical autism. These disorders are characterized by difficulties with interactions and social communication and a restrained range of interests and activities to those with a repetitive character [3]. ASDs begin in childhood and tend to persist into adolescence and adulthood. In most cases, ASD is identified in the first 5 years of life.

People with ASD often present other conditions, such as epilepsy, depression, anxiety, and attention deficit hyperactivity disorder (ADHD). The functional intelligence level of ASD persons is extremely variable, ranging from severe mental intellectual disability to very high IQ levels.

According to some nuclear magnetic resonance (NMR) studies [4], at the ages of 2–4 years, 90% of autistic patients have greater than average cerebral volumes.

Compared to the general population, autistic persons are at a higher risk of presenting with a range of concurrent medical disorders and premature mortality [5,6]. It has been observed that common genetic vulnerability and/or underlying biological mechanisms that involve more systems may contribute to a higher prevalence of somatic complications in autism [7,8].

Sensorial and motor deficits as well as fine and gross motor function disabilities are consistently reported in children with ASD and are correlated with the severity of the social communication deficiency [9,10]. Sensory abnormalities are often the earliest identifiable clinical features of developmental disorders [11,12]. Similar to social communication deficits, motor deficits may represent the basic features of autism when a larger spectrum of symptoms is taken into consideration [13].

Access to information from multiple sources provides many benefits, such as improving reaction times, the accuracy of identification, and processing efficiency. The ability to integrate visual and auditory stimuli can provide a base for social development, language, and communication.

In recent years, a high prevalence of sensory processing has been observed in autistic children that accounts for 30% to 100% of cases. For the first time, processing difficulties have been recognized as a diagnostic criterion for ASD. These difficulties have a negative impact on the development and learning capacities of the subjects as well as on their behavioral, cognitive, physical, and psychological functioning. This is the main reason behind the need to improve sensorial stimulation.

Many studies have shown a disturbance in the postural control efficiency of people with ASD with an increase in the disturbance of the center of pressure (CoP) parameters under two conditions: eyes open (EO) [14,15,16,17,18,19] and eyes closed (EC) [15,16,17].

Postural control during orthostatism also seems to be modified in children with ASD during difficult situations [15]. Postural control is based on the automatic system [15,20], involves paying attention during different simple or complex tasks, and provides postural control regulation [21].

Dual task (DT) is a paradigm that has been widely used to investigate the degree of automatic and attentional processing in postural control in children, as well as in older adults [14,21]. DT is capable of improving the capacity of a person to stand in a vertical position while it integrates afferents from all systems to reflect the degree of automatism in postural control [21]. DT contributes to observing the posture in a multimodal integration context, especially when combined with an associated cognitive task.

Skateboarding has a positive impact on the development of new motor abilities [22]. Scientists have shown the positive results of dancing on repetitive behavior, cognitive function, executive function [23], behavioral problems, physical fitness, and motor abilities [24,25] in children with ASD. The effect of gymnastic exercises on self-control was established in [15,26], and its effect on speech development and physical fitness indicators was shown in [26]. Exercise programs that involve cardio and fitness significantly increase fitness levels in children with ASD, improving aerobic resistance and muscle strength [27]. Exergaming use has been shown to reduce the number of stereotyped actions and improve cognitive and executive functions in children with ASD [23]. Outside games and training programs that mainly involve sport game elements increase PA [28] and positively affect motor abilities in children with ASD, including hand and body coordination, strength, and dexterity [29,30], as well as executive functions [31].

It has been demonstrated that many interventions could increase the balance capacities of children with ASD [29,30]. While many are non-specific (e.g., swimming or taekwondo [31]), we found only two studies that explored the possibility of using balance training in particular for the treatment of ASD [30].

Cheldavi H et al. [32] reported postural control improvement in two visual conditions (eye open and eye closed (EO, EC)) and on two surfaces (foam, hard) in 10-year-old children with ASD. In this study, 20 children with ASD undertook a training program based on a six-week-long balance training intervention with eyes open and eyes closed. The mean velocity and anteroposterior and mediolateral axis displacements were measured and they found that balance training improves the postural sway in different sensory conditions in children with ASD.

Travers [33] investigated the effect of a 6-week balance training program based on visual biofeedback in 29 children with ASD in EO, EC, and visual feedback conditions (the latter meaning they could see someone’s center of pressure on a screen). The results of the study show that specific balance training programs are capable of improving postural control and suggest that this should be integrated into rehabilitation programs for children with ASD [30].

At the same time, the use of physical activity for balance rehabilitation could help to improve posture and ASD phenotype, not only by releasing attentional resources but also by addressing one of the many possible causes of ASD symptoms.

Additionally, Jabouille et al. [34] found that 4 weeks of balance rehabilitation using specific sensory stimulation such as balance foam increased cognitive load conditions. The dual task conditions consisted of presenting images representing a neutral condition, sadness, anger, and fear. The evaluation included the assessment of postural control by measuring the surface, velocity, mediolateral, and anteroposterior sway amplitudes of centers of pressure using a posturographic platform. The rehabilitation program resulted in a 30–96% improvement in postural control parameters in both participants.

Research by Caldani et al. [35] on postural control in children with ASD during a specific rehabilitation program was conducted using Balance Quest by Framiral on an unstable platform under three different viewing conditions. The participants were split into two groups. Group G1 spent 1 min on postural training and group G2 spent 6 min on postural training. They concluded that G2 showed a significant improvement in postural control as assessed by the Framiral platform and that new rehabilitation strategies need to consider incorporating postural rehabilitation in children with ASD.

Sensory information processes from visual, vestibular, and proprioceptive receptors contribute to postural stability in order to accomplish neuromuscular control, balance maintenance, and appropriate motor responses. Any disorders involving these processes or integration disorders affect an individual’s balance, necessitating intervention. Ghafar et al. [36] propose an investigation into sensory integration and balance using the Biodex balance system (BBS) in children with autism spectrum disorder (ASD) during a static posture. They studied 74 children with ASD and evaluated their postural sway in four different situations: eyes open/firm surface, eyes closed/firm surface, eyes open/foam surface, and eyes closed/foam surface. Children with ASD showed a significant increase in postural sway and the results provide evidence that postural stability decreased in children with ASD. Additionally, they found that children diagnosed with ASD have postural control deficiencies, especially in cases in which visual and somatosensory input are disrupted. The authors conclude that there is a need to conduct more research to find the best balance training program using different rehabilitation exercises and to provide a scientific basis for a training program. Balance rehabilitation in particular seems to have the potential to improve postural control in children with ASD. For this reason, and due to the lack of information that is currently available, we proposed this study.

The aim of our study was to analyze the role of a multisensorial stimulation intervention in children with autism spectrum disorder (ASD) during postural control and balance exercises.

## 2. Materials and Methods

### 2.1. Subjects

We studied 28 participants aged between 8 and 14 years old. Their average age was 8.6 years and their average weight was 32.18 kg with a standard deviation (SD) of 8.22. All participants had autism or autism spectrum disorder (ASD). They did not present aggressive behavior, intellectual disability, or ADHD. All subjects had similar levels of development, meaning that there were no significant differences between those who were 8 years old and those who were 14 years old. In Table 1, we present the demographic data. Figure 1 presents the gender distribution of the patients.

ASD is a genetically heterogenous group of neurobehavioral disorders characterized by impairment in three behavioral domains including communication, social interaction, and stereotypic repetitive behaviors. For this reason, it is possible to have different responses to tasks and training. We tried to select a homogenous group in order to ensure that specific aspects such as communication, social interaction, and stereotypic repetitive behaviors did not influence their participation in the training. However, all participants recruited in the present study had level one autism (mild) and were able to complete a series of motor competence assessments (i.e., Bruininks–Oseretsky Test of Motor Proficiency-2 and Movement Assessment Battery-2). The results of these assessments are not presented because this was not the aim of our research. We enrolled patients according to their medical and social assistance requirements. Children with level one ASD exhibit deficits in social communication without supports in place and inflexibility of behavior. Inclusion criteria: (1) diagnosed with level one ASD by a physician, (2) had the ability to understand and communicate with the physician and physiotherapist, and (3) had the ability to perform motor tasks. The exclusion criteria: had chronic medical disorders, visual impairments, physical impairments that could affect postural stability, attention deficit hyperactivity disorder (ADHD), intellectual disability, and had not participated in any physiotherapy programs to improve balance.

The participants are included in a special education system. The children attend a special education center and are diagnosed with ASD from authorized clinical personnel before they come to us to work on the rehabilitation of their balance and coordination. We receive their medical documents, which contain only diagnoses based on medical decisions. The grade of severity is assessed based on medical diagnosis, medical evaluation, and special criteria such as: biopsychosocial assessments, participation in social activities, physical activity, participation in the community, learning skills, selfcare, communication, and language mobility.

According to specific scales, children with ASD are evaluated using the Vineland scale (average 8.6/36 points); Harvey scale (average 29.75/93 points); and psychometric development scale (average value is 96.5%).

We excluded children with ADHD and intellectual disabilities because they do not show sufficient cooperation when participating in physiotherapy programs. The aim of our study was to increase participants’ balance and coordination and we worked with children that had the capacity to understand and respond to the tasks in the training program.

### 2.2. Evaluation

The evaluation was conducted using the RSScan pressure and force platform to register the parameters, including the surface of the confidence ellipse (A) and the length of the curve (Lc) described by the pressure center and coefficient Lc/A. These parameters were registered before and after the physical exercise program in two evaluations: one before (EV1) and one after (EV2) the physical exercise (P.E.) program, which lasted 6 months.

The assessments were conducted in a quiet room with constant light and temperature levels. The children had to be barefoot during the assessment. The children had to stay calm on the platform while looking at an empty white wall. In this way, we avoided any type of distraction, and we used simple instructions such as “stay!” and “do not move!”.

Posturographic data were obtained using the RSScan platform. The high-speed system performs accurate plantar pressure measurements with 4096 sensors at a scan rate of up to 500 Hz or 500 measurements per second. Raw data were filtered offline before counting the CoP parameters: the surface (the ellipse that contains 90% of the CoP coordinates), medium speed (medium speed of the CoP during the acquisition period, which means 30 s), and the CoP in the anteroposterior interval (AP) and mediolateral directions (ML) [16,17]. The CoP is the center of pressure in terms of body weight distribution, as recorded by force and pressure platforms.

The posturographic assessment is reliable and valid in children with typical development [17].

### 2.3. Statistical Analysis

The statistical analysis included descriptive analysis and a JB test (Jarque–Bera), which gives us information about the normal distribution of parameters. A Student’s t test was applied to reveal any differences between parameter values from evaluation point 1 (EV1) to evaluation point 2 (EV2). The test indicates whether it is a significant difference.

We applied the Student’s *t*-test for equal means. To analyze the effect size of the parameter evolution, we used the Cohen D coefficient. For correlations between parameters at the two points of evaluation—EV1 and EV2—we use Pearson’s correlation and Spearman’s correlation.

### 2.4. Training Program—Physical Activity (P.A.)

For therapy recommendations, based on the results of the evaluation, we proposed an initial P.A. therapy program based on a series of principles. We worked with each participant individually 3 times per week for 6 months using the program listed below. Each session lasted 30 min and followed the list of 6 exercises that we recommended in the exact order they are listed, from 1 to 7, starting with warm-up exercises.

We made our recommendations based on the evaluations performed with child interventions involving physical exercises for reducing the imbalances specific to autism, such as behavioral problems, stereotypical movements, lack of attention (same as for improving academic performance), social involvement, relations between peers, and motor perceptive skills.

Breathing exercises and stabilization exercises were used to restore muscle posture balance and the vertical axis of the body (in the same way as gymnastics), and the entire program was designed around sensory–motor coordination development, which is the most important step in forming the “cognitive–emotional brain” and intellectual abilities.

The objectives of the kinetic program were as follows:Positive influence on the body scheme representation;Recovery of laterality disturbances;Recovery of orientation, organizing, and spatial structure problems;Recovery of orientation and temporal structuring disorders;Recovery of balance and coordination disorders.

The modalities that are helpful towards achieving these objectives are the physical exercises that are proposed below.

Exercises that are useful for increasing both-leg and one leg balance as well as for improving coordination and stabilization were included. We recommend performing them 3 times per week.

Exercises and games were organized using simple equipment such as gymnastic benches and balance boards. For a complex and detailed rehabilitation program, we recommend the following:

Warm-up exercises for the neck and head (flexion–extensions), upper limbs (flexion–extensions, abductions, adductions, rotations), trunk (lateral left–right inclinations, trunk rotations), and lower limbs (flexion–extensions, circumductions, ball-like jumping and tip–toes–heels lifting) were recommended for the beginning of every training session.

We recommend patients walk on the wide side of a gymnastic bench (see Figure 2) for 4 repetitions 3 times per week. (This exercise is useful for increasing the balance and coordination of the lower limbs.)
Walking on a gymnastic bench, on the narrow side, as seen in Figure 3. (This exercise is useful for increasing the balance and coordination of the lower limbs; using the upper limbs to maintain balance means that we train those too.) We recommend that patients perform 4 repetitions of this exercise, 3 times per week.Standing on one leg, like a stork, for 10 s per leg (this exercise improves balance and coordination). We recommend that patients perform 6 repetitions of this exercise, 3 times per week.Standing on one leg, with the other leg in front, lateral, and behind without touching the ground (Figure 4). (This exercise is good for increasing dynamic balance.) We recommend that patients perform 6 repetitions of this exercise, 3 times per week.Jumping from one leg to another while using circles on the ground to mark the place where the child must jump. (This exercise improves dynamic balance and coordination.) We recommend that patients perform 4 repetitions of this exercise, 3 times per week.Balance board exercises. First, we need to help the child and keep him or her balanced by placing the board near a parallel bar, where they can maintain their own balance. When the child is confident enough, he or she can release the support and stay on the board by themselves (as shown in Figure 5). This should be performed 3 times per week. (This exercise is useful for increasing balance.)

## 3. Results

The initial evaluation results are presented in Table 2 and the descriptive statistics of the evaluation results are shown in Table 3.

A final evaluation (EV2) was conducted after 6 months, and the raw data are presented in Table 4. The descriptive statistics of the evaluation results are presented in Table 5.

The Jarque–Bera (JB) test provides us information about the normal distributions of parameters. We can see that we have a normal distribution for all parameters. For this reason, the Student’s t test could be applied.

We applied the Student’s t test for equal means. The Student’s t test reveals whether there is any difference between the parameter values between EV1 and EV2 and whether these differences are significant. To analyze the effect size of parameter evolution, we used Cohen’s D coefficient. The results of the Student’s t test and Cohen’s D test (coefficient) are presented in Table 6.

In Table 6, we can see that the Student’s t tests showed us *p* values that did not meet the significance value of 0.05. Thus, the alternative hypothesis is accepted, meaning that the values of the surface of the confidence ellipse, the length of the curve described by the COP, and the coefficient Lc/A shows a significant improvement at EV2 compared with EV1.

For correlations between parameters at EV1 and EV2, we used Pearson’s correlation and Spearman’s correlation and the results are presented in Table 7 and Table 8.

Our results show the following:

The *p* values are less than the significance value and this means that the values for all 28 participants show significant differences between EV1 and EV2.

The average values of the surface of the confidence ellipse shows a decrease of 92%, from an average value of 927.32 mm^2^ to 67.91 mm^2^, and a decrease of 42% can be seen in terms of the length of the curve described by the COP (decreased from 644.44 mm^2^ to 374.67 mm^2^). The standard deviation decreased between EV1 and EV2, and this means that the values are more grouped around the mean value. We also observed a good correlation between the evolution of the surface of the confidence ellipse at EV1 and EV2 (coef Person = 0.911; coef Spearman = 0.637), but low correlations between the other parameters.

## 4. Discussion

The results of our research respond to the challenge regarding the improvement of balance and stability in people who suffer from ASD using multisensory stimulation. Positive effects were observed under the influence of exercises that required complex coordination and balance rehabilitation regarding improvements in postural control. According to [37] and Moseley and Pulvermüller [38], structural changes in ASD patients’ brains may lead to a number of subtle deficits in motor control, including postural instability, which may eventually interfere with social and cognitive development by reducing opportunities to explore and interact with people and the environment. Brain plasticity is the main property that is involved [38,39]. Imbalance rehabilitation may focus therapeutically on early signs of central nervous system (CNS) anomalies as they appear in motor dysfunction, consequently attenuating social and cognitive deficits. For this reason, our proposal program influences the body scheme representation and involves restoring and improving brain activity in the field of motor control.

Morris and collaborators, in their 2015 study [18], speak about multi-sensorial integration, which is poor in children with ASD. Additionally, it has been observed that postural control in children with ASD is more affected by DT emotions and interaction than in typical development children [30,32]. Despite this, researchers report a higher CoP speed and a higher CoP surface during social image observations (meaning observing faces) than in images that contain neutral objects in children with ASD compared to a control group. Another study reported similar results by comparing balance performance in 30 children with ASD (12.1 ± 2.9 years old diagnosed by ADI-R, ADOS and DSM-5 criteria) with healthy children with average IQs when they were shown different emotional images of the face [30]. Impaired postural stability in children with ASD is well established in the literature [40], and the use of force plates by other authors demonstrate that it is also important to analyze the CoP. Children with ASD present greater CoP sway displacements [41], sway areas [42], standard deviations of COP coordinates [42], sway velocities [43], and root mean square of COP coordinates [44]. Li Y [45] investigated age’s effect on postural stability in children with ASD in terms of comparing the amplitude and complexity of CoP sway during quiet standing in children with ASD among three different age groups: 6–8 years (under 8: U8), 9–11 years (U11), and 12–14 years (U14). The author found that the U14 group exhibited improved mediolateral postural stability compared to the U8 group, whereas no differences were found between the U8 and U11 or between the U11 and U14 groups. This study demonstrates that children with ASD have the possibility to develop postural stability slowly, but they observed significant changes over a long period of time. This is in accordance with our observations about the importance of introducing multisensorial training early to develop postural stability and increase balance and coordination.

We recommended early intervention programs specifically focused on improving the complexity of postural control as potentially beneficial for children with ASD. Future research is warranted to investigate postural control complexity in young autistic children and to evaluate the efficacy of early interventions at enhancing postural stability.

Many authors suggest that, due to increased cognitive challenges, children with ASD that present postural control problems are less capable of integrating emotional social cues while they are standing than unaffected children. The conclusions of this study demonstrate the importance of administering balance training combined with different sensorial stimuli to children with ASD.

Additionally, the training program significantly improved inferior limb strength, in contrast to superior limb strength, which presented no obvious improvements. This result may be explained by the training program that was used. In all training sessions, the exercises and the proposed content involved different types of static and dynamic balance, which made greater demands of inferior limbs. We should note that the upper limbs were used to maintain balance and the correct body posture, and this is in accordance with Laurenco’s results [46].

Our results are also in accordance with those presented by Cheldavi et al. [32], which found that balance training (45 min/session; 3 sessions per week) improved postural control, which was assessed using a Bertec force plate. In terms of the training itself, our program is in accordance with the results of Najafabadi et al. [47], which proposes Sports, Play, and Active Recreation for Kids (SPARK). Their proposal involves 36 sessions (3 sessions per week; 40 min per session), and the results are improved balance (static and dynamic), bilateral coordination, and social interaction.

In terms of innovation and significance, we consider research about postural sway to be complex in children with ASD, which necessitates the monitoring of the evolution of balance and postural control because the development of postural control is a dynamic process through which children learn to control multiple degrees of freedom of body segments to maintain balance. It is an innovation because we try to present an analysis of postural control in response to a specific training because, in recent years, many studies have investigated CoP complexity during quiet standing tasks in different populations, but not in children with ASD. Additionally, postural sway complexity can reflect the adaptive capacity of the postural control system, which is the result after this specific training regimen.

The limitations of this study include having a small number of children involved in the research, sometimes inconsistent participation in the training program, and poor communication with the physiotherapist. Because we had a small sample size, we had no control group, partly because only a small number of children and their parents consented to participate in the study, and also due to their inconsistent participation.

## 5. Conclusions

Future research is needed to test specific types of physical training in children with ASD due to the complexity of their responses.

Specific motor intervention that comprises different types of physical exercises and materials and that uses ludic exercises may potentially be a more effective strategy.

To improve patients’ balance, postural control, and motor skills, we need a program whose aim is to work on fundamental motor skills based on multisensorial stimulation.

Specific types of physical exercise in this population improve their physical condition, cognitive and sensorial capacities, motor performance, and motor coordination (gait, balance, arm functions, and movement planning).

Physical exercise based on multisensory stimulation is a useful tool for the development of children with ASD and is becoming increasingly used, but we need to conduct evidence-based scientific research that supports this practice, giving it greater scientific robustness.

The motor intervention programs we propose for children with ASD provide benefits in many different domains.

## Figures and Tables

**Figure 1 children-09-01152-f001:**
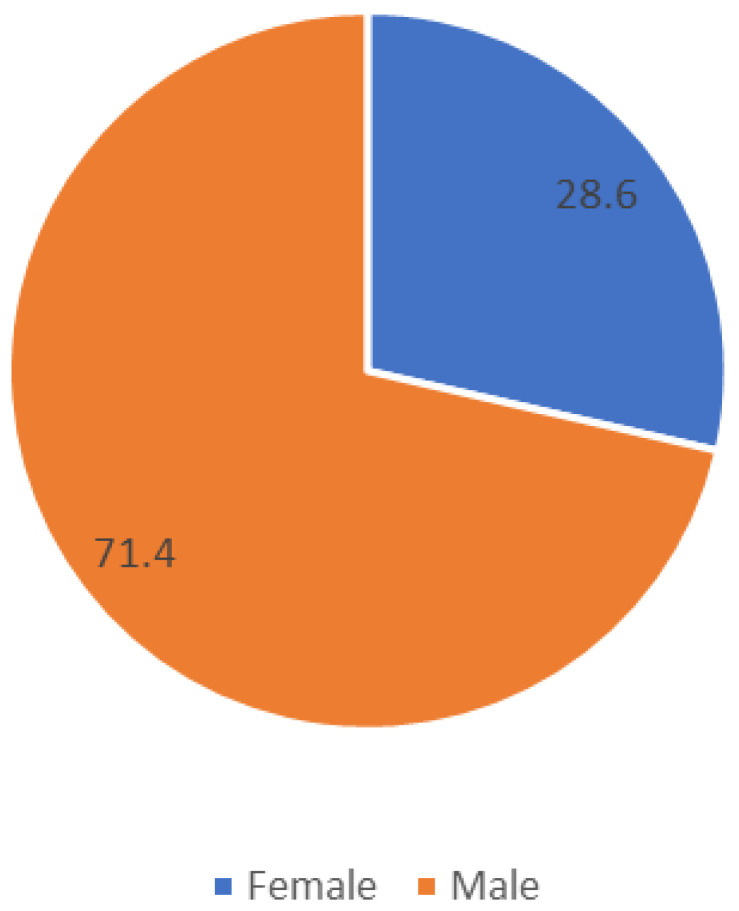
Gender distribution of patients. Participants are included in the special education system.

**Figure 2 children-09-01152-f002:**
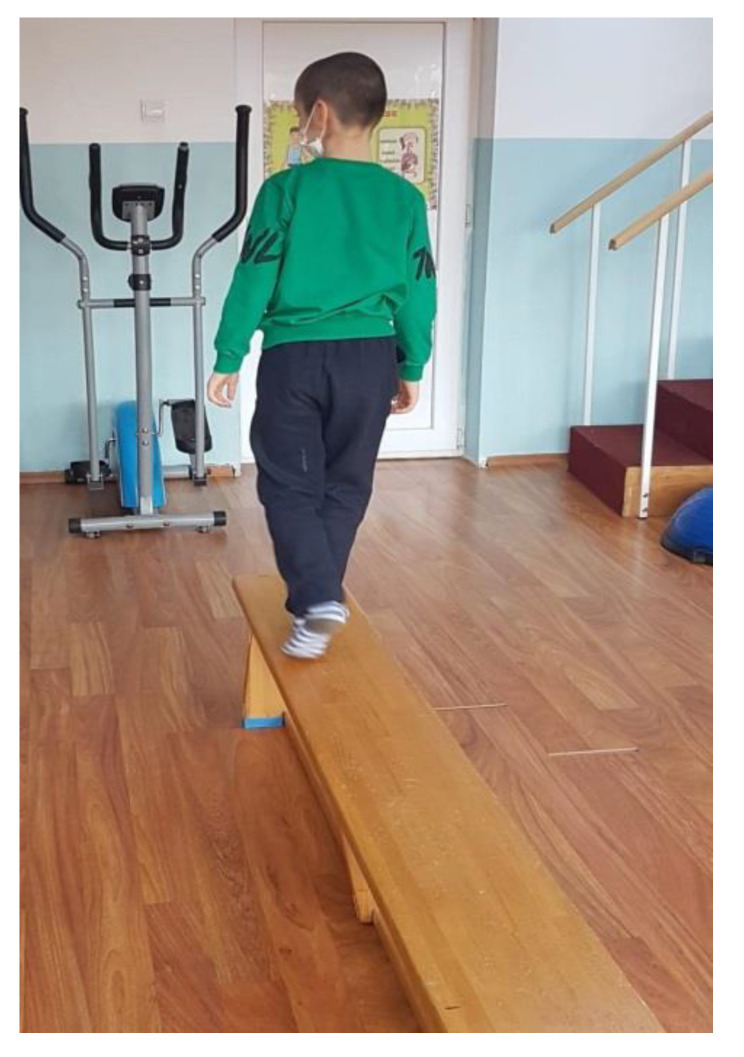
Walking on the wide side of a gymnastic bench.

**Figure 3 children-09-01152-f003:**
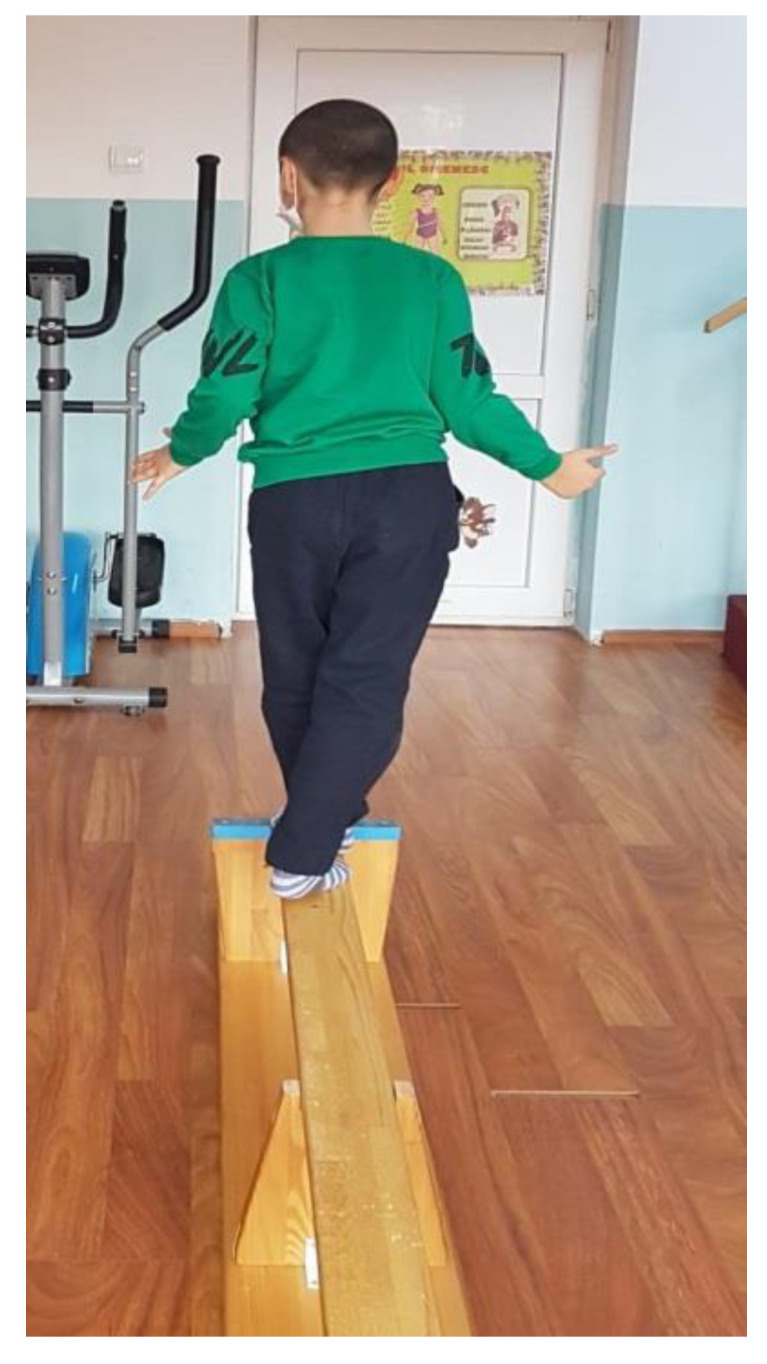
Walking on the narrow side of a gymnastic bench.

**Figure 4 children-09-01152-f004:**
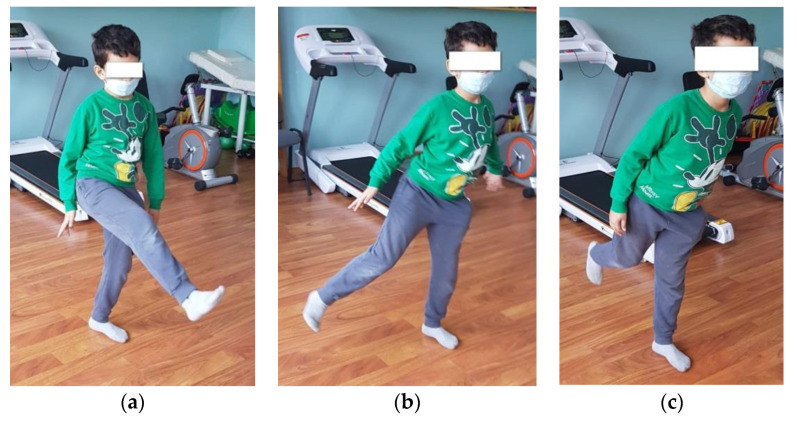
Standing on one leg while the other is in front (**a**), lateral (**b**), and behind (**c**).

**Figure 5 children-09-01152-f005:**
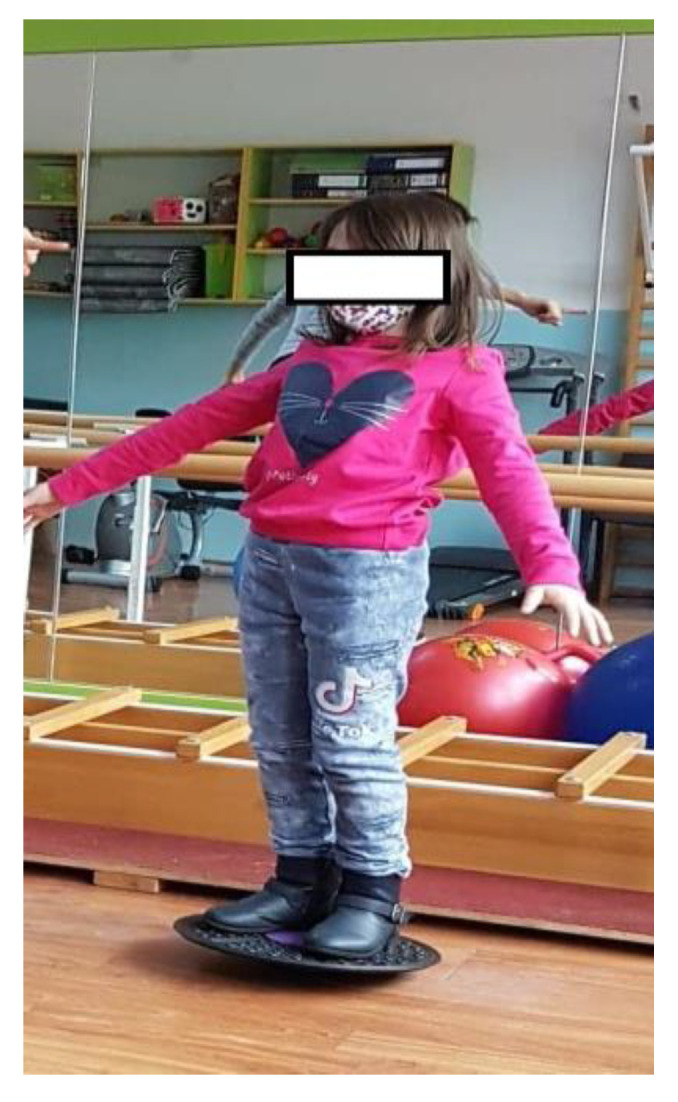
An image representing the exercise on the balance board.

**Table 1 children-09-01152-t001:** Demographic data for subjects included in the study.

Subject No	Weight (kg)	Height (cm)	Gender	Level of Disability According to the Social Assistance Criteria
S1	37	134	F	Mild
S2	27	124	M	Mild
S3	36	134	M	Mild
S4	18	126	F	Mild
S5	22	125	M	Mild
S6	44	147	F	Mild
S7	44	138	M	Mild
S8	38	145	F	Mild
S9	25	128	M	Mild
S10	35	136	M	Mild
S11	20	137	M	Mild
S12	23	128	M	Mild
S13	45	150	F	Mild
S14	42	136	M	Mild
S15	40	140	F	Mild
S16	26	129	M	Mild
S17	37	136	M	Mild
S18	21	128	M	Mild
S19	24	124	M	Mild
S20	47	149	M	Mild
S21	41	151	M	Mild
S22	30	147	F	Mild
S23	26	132	M	Mild
S24	34	150	M	Mild
S25	17	135	M	Mild
S26	20	152	M	Mild
S27	43	142	F	Mild
S28	39	129	M	Mild

**Table 2 children-09-01152-t002:** Data collected from subjects at the first assessment (EV1).

Subjects	Weight (Kg)	The Surface of the Confidence Ellipse A (mm^2^)	The Length of the Curve (Lc) Described by COP (mm)	Coefficient Lc/A
S1	37	326	787	2.41
S2	27	357	525	1.47
S3	36	1489	407	0.27
S4	18	1437	900	0.63
S5	22	2935	1144	0.12
S6	44	30	480	16.16
S7	44	30	297	10.05
S8	38	328	788	2.40
S9	25	356	527	1.48
S10	35	1490	410	0.28
S11	20	1440	910	0.63
S12	23	2740	1150	0.12
S13	45	30	481	16.03
S14	42	29	300	10.34
S15	40	322	780	2.42
S16	26	353	520	1.47
S17	37	1480	417	0.28
S18	21	1430	912	0.64
S19	24	2816	1104	0.11
S20	47	30	470	15.77
S21	41	27	279	10.51
S22	30	336	786	2.34
S23	26	367	523	1.43
S24	34	1469	415	0.28
S25	17	1487	900	0.61
S26	20	2773	1094	0.11
S27	43	32	460	14.51
S28	39	28	279	10.13

COP =center of pressure.

**Table 3 children-09-01152-t003:** Min, max, and mean value and SD at the first evaluation.

	Weight (Kg)	The Surface of the Confidence Ellipse A (mm^2^)	The Length of the Curve (Lc) Described by COP (mm)	Coefficient Lc/A
Minimum	17	26.55	279	0.11
Maximum	47	2935	1150	16.16
Mean	32.18	927.32	644.44	4.39
Standard deviation	9.50	977.42	283.43	5.75

**Table 4 children-09-01152-t004:** Data collected from subjects at the final assessment (EV2).

Subjects	Weight (Kg)	The Surface of the Confidence Ellipse A (mm^2^)	The Length of the Curve (Lc) Described by COP (mm)	Coefficient Lc/A
S1	38	18	398	22.16
S2	28	3	268	89.77
S3	38	125	385	3.06
S4	21	37	274	7.33
S5	24	246	620	2.52
S6	43	18	351	19.25
S7	46	21	344	16.18
S8	33	20	391	19.58
S9	30	5	277	56.65
S10	40	127	384	3.01
S11	23	38	269	7.15
S12	28	240	617	2.57
S13	46	19	341	17.67
S14	42	19	334	17.34
S15	32	18	388	21.60
S16	25	4	277	75.22
S17	35	135	389	2.87
S18	26	35	249	7.04
S19	27	236	618	2.62
S20	43	18	349	19.25
S21	45	27	350	12.83
S22	39	17	388	22.87
S23	29	3	269	99.67
S24	34	129	371	2.88
S25	18	36	280	7.70
S26	22	264	613	2.32
S27	36	19	353	18.38
S28	37	22	349	16.04

**Table 5 children-09-01152-t005:** Min, max, and mean value and SD at the final evaluation.

	Weight (Kg)	The Surface of the Confidence Ellipse A (mm^2^)	The Length of the Curve (L) Described by COP (mm)	Coefficient Lc/A
Minimum	18	2.70	249	2.32
Maximum	46	264	619.50	99.67
Mean	33.14	67.91	374.67	21.27
Standard deviation	8.22	84.23	110.66	26.32

**Table 6 children-09-01152-t006:** Student *t* test and Cohen’s D coefficient values for parameters collected at EV1 and EV2.

	The Surface of the Confidence Ellipse	The Length of the Curve Described by the COP	Coefficient Lc/A
*p* values (results of Student’s *t* test)	0.004	0.000	0.0016
Cohen’s D test	0.8	1.25	−0.9

Significance level *p* = 0.05.

**Table 7 children-09-01152-t007:** The coefficients of Pearson correlation for the values of parameters collected at EV1 and EV2.

Variables	EV2 Weight	EV2 The Surface of the Confidence Ellipse	EV2 The Length of the Curve Described by the COP	EV2 Coefficient Lc/A
EV1 Weight	0.918	−0.374	−0.128	−0.079
EV1 The surface of the confidence ellipse	−0.620	0.911	0.693	−0.456
EV1 The length of the curve described by the COP	−0.721	0.548	0.549	−0.258

Significance level *p* = 0.05. EV1 = evaluation moment 1; EV2 = evaluation moment 2.

**Table 8 children-09-01152-t008:** The coefficients of Spearman correlation for the values of parameters collected at EV1 and EV2.

Variables	EV2 Weight	EV2 The Surface of the Confidence Ellipse	EV2 The Length of the Curve Described by the COP	EV2 Coefficient Lc/A
EV1 Weight	0.903	−0.374	0.094	0.352
EV1 The surface of the confidence ellipse	−0.673	0.637	0.321	−0.616
EV1 The length of the curve described by the COP	−0.746	0.246	0.199	−0.215

Significance level *p* = 0.05.

## Data Availability

Not applicable.

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
