# Peer review of "Physical Activity Design for Balance Rehabilitation in Children with Autism Spectrum Disorder"

_children, 2022, doi:10.3390/children9081152_

Round 1
Reviewer 1 Report
The aim of this study is to analyze in 28 ASD children the postural stability evolution after the physical therapy exercises based on balance training.
- There is a significant amount of literature about epidemiological studies in ASD identifying multiple pathophysiological relevant phenotype features present in some, but not all ASD subjects. Considering this, most of the facts reported in the introduction and section “1.1.Causes, clinical aspects of autism/ASD cruelly lack of references. Please add where appropriate to support the statements you are making.
- The recruitment criteria are not clear at all. According to what criteria was a diagnosis of ASD made?? Which multidisciplinary team carried out the clinical assessments? No data regarding individuals' cognitive and social ability level is reported. And according to what assessment children with ADHD or intellectual disability were excluded? Moreover, the authors write "mental retardation" which is an obsolete medical terminology. I was also very curious to know how many individuals were excluded from the potential population because of the exclusion criteria. A flowchart diagram of subject’s selection is needed as well as a table with demographic data.
- Congenital genetic syndromes are increasingly implicated in ASD and these syndromes are often associated with neurosensory deficits and motor/functional impairment. Children with ASD were genetically assessed? If not the authors must to include in the study the implications that this may have.
- It is stated in literature that anomalies in postural stability and muscle tone are directly associated with autism severity. No data was available about severity of autism in this study. I think autism severity evaluation in ASD group is needed taking into account the implications that autism severity data may have in the results.
- Tables of the results are not clear at all.
- The discussion is poorly written and mainly focused on the consistency of the current research findings with previous ones. There is no text focused on significance and innovation. I do not understand why the authors claimed that their result is consistent with previous ones given.
- Why the authors did not explain which the study’s limitations are?
- The acronym for autism spectrum disorders has been internationally known as “ASD” for decades… Which is the meaning for DSA?
Author Response
Dear reviewer,
Thank you very much comments and sugestions. We try to answer point by point and hope that answer to all.

Reviewer 2 Report
Physical activity design for balance rehabilitation in children with autism is trying to evaluate the impacts of a physical intervention approach on the posture and balance improvement of individuals with ASD. An interesting topic.
I suggest using one specific term to describe individuals with ASD. Even in the title, it is recommended to use the presently accepted term of autism spectrum disorder instead of autism.
The language in terms of the tens needs a revision. I think the used tens in this part must be the past tense. When interpreting the results or describing the significance of the findings, the present tense should be used. Still, in the discussion section, the past tense is generally used to summarize the results.
In the introduction:
The used style for writing decimals (0,3%) is unfamiliar (I mean using a comma instead of a full-stop). I suggest using the APA style for reporting (0.3%).
Material and Method:
What RSScan stand for? ADHD? It is recommended that all the abbreviated terms be presented fully at the beginning, along with an abbreviation in parenthesis.
1. I suggest substituting intellectual disability instead of mental retardation, which has been used to address individuals with intellectual issues for at least two decades.
2. I also think that providing more information about the sample is crucial.
3. How did the sampling approach implant?
4. How many individuals were excluded because of the exclusion criteria?
5. How did the 28 individuals in your sample receive a diagnosis?
6. What scale was used, how did the diagnosis, and make it clear what do you mean by Autism and Autism Spectrum Disorders?
7. How did children at 8 be similar regarding the development to the adolescent of 14?
8. How do you know that they had no other developmental issues?
9. What do you mean by not essential differences? (You mean no statistically significant difference? If so, you need to report numerical data.
Findings:
The reported findings (Table 5) (Table 6, and Table 7) are not according to the APA style of presenting results, which is unfamiliar to me.
Significant correlation levels need to be marked
I did not understand why the statistically significant levels are reported in the following style: (Line 254)
(p values are less the significance value and this means that the values for all 28people have a significant difference between EV1 and EV2.)
I also recommend that 28 sample members be substituted with 28people
Significant needs to be substituted with significance
Discussion
I think “higher CoP” as a term needs more explanation and what the abbreviated part stands for.
Typically developing is a more acceptable substitution for healthy children (line 283).
I suggest children with ASD be substituted with (ASD children).
I suggest re-wording and reorganizing the final part of the discussion (lines 297-303) from both the format and presentation.
There are obvious limitations with this study, such as the lack o diagnostic data, having no control group, using only a pre and post-design, small sample size, and many other points that deserve to be presented in a particular part.
I also suggest reorganizing the (Conclusion) part.
I also suggest the paper be proofread, especially the final part.
Author Response
Dear reviewer,
Again thank you for effort and helping us to improve our paper.

Reviewer 3 Report
Autistic Spectral Disorders are a range of neuro-developmental disorders that include autism. The characteristics of ASD include deficits in cognitive processing, impaired social interactions, delayed or limited communication skills, and restrictive patterns of activities or interests. This manuscript is important to find a way and possibilities to help kids with ASD.
Can you explain how this text is correlates this your research?
So, the researchers found a positive impact of skateboarding over forming of the 101 new motor abilities [20]. Scientist proved positive results of dancing over repetitive behavior, cognitive function and executive function [21], behavioral problems, physical 103 fitness and motor abilities [22], [23] of the ASD children. The effect of gymnastic exercises 104 over self-control was established [12], [24] and over speech development and physical 105 fitness indicators [24]. Exercises programs that involve cardio and fitness significantly 106 increased fitness level in ASD children, they firstly improved aerobic resistance and 107 muscle strengths [25]. Exergaming use reduced the number of stereotyped actions, im- 108 proved cognitive and executive functions in ASD children [21]. Outside games and 109 training programs that use mainly sport games elements increased PA [26] and positively 110 affected motor abilities in ASD children: hand and body coordination, strengths and 111 dexterity [27], [26], and executive function [27].
Line 131
Is not used in international citation kilos
Discussion section is described very succinctly. It's not clear enough, but what you're doing with your research is new.
Author Response
Dear reviewer,
Thank you very much for comments that help us to improve the paper quality.

Reviewer 4 Report
I would like to thank the authors for taking the time to conduct this research and writing this manuscript physical activity design for balance rehabilitation in children with autism. Even though I find this a very interesting topic, I believe that the manuscript needs to be reviewed to make improve its scientific soundness. I would like to acknowledge the hard work I am sure all authors put into conducting this study and writing this manuscript; however, the language use throughout the text affects the effective communication of ideas and makes the research somewhat difficult to understand. I would advise authors to seek professional English language guidance, as the grammar and fluency could significantly be improved. The writing mechanics should be reviewed as well, as it could significantly improve the organization and development of ideas throughout the manuscript. I would suggest paying close attention to verb tense (subject-verb agreement), sentence structure, punctuation marks, and word choice. I would also suggest that authors provide more background information in the introduction, so the reader can contextualise the problem, and thus understand the relevancy of this work. Also, I would suggest reviewing the references used for this manuscript as in some instances it feels the references are missing or not cited. In short, I would suggest reviewing the structure of the manuscript (introduction, method, , analysis, results, and discussion).
Author Response
Thank you very much for the time spent for evaluate our paper and thank you for your recommandation. We try to answer and hope in good feedback from you.

Round 2
Reviewer 1 Report
After the authors' changes, the scientific sound of the study is improved. I accept the article in the presente form
Author Response
Thank you very much for your effort and comments.
Reviewer 2 Report
It is an updated version of the previously submitted paper. Hence, there are some issues to resolve. First of all, it is said that the medical doctors or physicians made the diagnosis. I suggest replacing the authorized clinical personnel instead with medical doctors. The reason is that Autism spectrum disorders are not a purely medical condition and to be able to make the diagnosis, extra training is needed for all the rehabilitation and health personnel engaged in development can apply for participation.
The second point is that ASD children are still used in the entire text, and I suggested to be replaced with children with ASD. I also noticed that you agreed with my suggestion and accepted the substitution in your comments; hence, it is still left unchanged in the text (22 times!).
There were many typos and issues with the applied tense, and I am sure English language editing and final proofreading by a negative English speaker are essential. As an example line, 296 started with (In Tabel) that needs to be (In table).
Author Response
Thank you very much for your sugesstions and for help us to improve the paper quality. We hope will be a postive feedback from you and thank you for this.

Reviewer 4 Report
Once again, I would like to thank the authors for all the efforts they have put into this manuscript and for reviewing it considering all the feedback provided. Nonetheless, I still have concerns about the scientific soundness of this work and continue to believe that the manuscript needs to be thoroughly reviewed. I think that it would be advisable for authors to seek professional English language guidance, or have a native English speaker go over the manuscript. Throughout the text the language use is affecting how the information is conveyed. The grammar and fluency could significantly be improved as the writing mechanics fails to properly address the ideas, which are essential to understand the work the authors have carried out. I am still missing more contextualization of the topic in the introduction. As one reads the text it feels that it lacks the empirical information needed to understand the objectives of this work, and organization of ideas makes the text somewhat difficult to read, as there are many language-use and grammar aspects affecting the fluency of the text. It is important to support the proposed ideas with empirical research, as authors provide statistics, and it is not quite clear where they have been taken that information from. For that reason, I strongly suggest thatauthors provide more background information in the introduction, so the reader can contextualize the importance of the study conducted, and thus understand the relevancy of this work. The references used for this manuscript should also be reviewed, as in some instances it feels the references are either missing or not cited in the correct place.
Here are some specific concerns I have about the manuscript:
My first concern is the length of the introduction and the organization of this section. I am missing more empirical support of the ideas expressed throughout the text.
Another question I have is about where the introduction ends and why authors have chosen to use subheadings such as 1.1 causes, clinical aspects of autism/ASD. Is this section still part of the introduction? This is somewhat confusing, and thus I would suggest going over the structure given to this section.
In lines 139-141 language use is affecting the clarity of the study’s objectives, I would suggest to authors to please review the grammar as clearly stating the study’s objectives is crucial.
I think authors need to be clear in the writing style the manuscript is written, as citations seem to be cited in Vancouver, when the most common style for these types of manuscripts is APA.
Please be mindful of how statistical data is reported. For instance, in Materials and Methods, there is no consistency in the terminology used.
I would suggest restructuring the sections Materials and Methods (line 143), for instance, having methods as the main heading and then subheadings for subjects, materials, evaluation, and so on. This way the reader will be able to coherently follow the text.
I would also suggest authors review the format and information provided in table 1 (line 188-189), as it is somewhat hard to read.
Author Response
Thank you very much for your comments and time spent for help us to improve the paper quality.

Round 3
Reviewer 2 Report
The comments and suggestions are considered by the authors.
Author Response
Thank you very much for time spent for read our paper and also for your comments that help us for improve the paper quality.